

# Spatial benthic community analysis of shallow coral reefs to support coastal management in Culebra Island, Puerto Rico

Nicolás X. Gómez-Andújar[1,2,3] and Edwin A. Hernandez-Delgado[1,2,4]

[1] Department of Environmental Sciences, University of Puerto Rico, Río Piedras Campus, San Juan, Puerto Rico
[2] Sociedad Ambiente Marino, San Juan, Puerto Rico
[3] Marine Resource Management, College of Earth, Ocean and Atmospheric Sciences, Oregon State University, Corvallis, OR, USA
[4] Center for Applied Tropical Ecology and Conservation, University of Puerto Rico, San Juan, Puerto Rico

Corresponding authors
Nicolás X. Gómez-Andújar,
gomezann@oregonstate.edu
Edwin A. Hernandez-Delgado,
edwin.hernandezdelgado@gmail.com

## ABSTRACT

Caribbean coral reefs provide essential ecosystem services to society, including fisheries, tourism and shoreline protection from coastal erosion. However, these reefs are also exhibiting major declining trends, leading to the evolution of novel ecosystems dominated by non-reef building taxa, with potentially altered ecological functions. In the search for effective management strategies, this study characterized coral reefs in front of a touristic beach which provides economic benefits to the surrounding coastal communities yet faces increasing anthropogenic pressures and conservation challenges. Haphazard photo-transects were used to address spatial variation patterns in the reef's benthic community structure in eight locations. Statistically significant differences were found with increasing distance from the shoreline, reef rugosity, *Diadema antillarum* density, among reef locations, and as a function of recreational use. Nearshore reefs reflected higher percent macroalgal cover, likely due to increased exposure from both recreational activities and nearby unsustainable land-use practices. However, nearshore reefs still support a high abundance of the endangered reef-building coral *Orbicella annularis*, highlighting the need to conserve these natural shoreline protectors. There is an opportunity for local stakeholders and regulatory institutions to collaboratively implement sea-urchin propagation, restoration of endangered Acroporid coral populations, and zoning of recreational densities across reefs. Our results illustrate vulnerable reef hotspots where these management interventions are needed and recommend guidelines to address them.

## INTRODUCTION

Coral reefs are biodiverse and socio-economically important ecosystems (*Connell, 1978*; *Moberg & Folke, 1999*), yet have suffered a global decline in recent decades, leading to the

loss of ecological functions, ecosystem resilience and services to humans across multiple spatial scales (*Hughes, 1994*; *Burke et al., 2004*; *Micheli et al., 2014*; *Maynard et al., 2015*). Sea surface warming and ocean acidification pose some of the most significant threats to coral reefs (*Eakin et al., 2010*; *Hughes et al., 2013*; *Hoegh-Guldberg et al., 2017*). An estimated 94% of corals have already experienced at least one episode of severe bleaching since 1980 due to extreme ocean temperatures driven by climate-change (*Hughes et al., 2018*). Additionally, more direct anthropogenic impacts, such as physical harm, overfishing, and land-based pollutants, continue to play a role in coral disease and mortality (*Ramos-Scharrón, Hernández-Delgado & Amador, 2012*; *Lapointe et al., 2019*). Local stressors, such as sedimentation and physical damage to colonies from recreational activities, have been recognized as a cause to the deterioration of coral reefs in the Caribbean Sea, specifically to reefs in Puerto Rico (PR) (*Jackson et al., 2014*; *Ramos-Scharrón, Torres-Pulliza & Hernández-Delgado, 2015*; *Webler & Jakubowski, 2016*). Furthermore, some areas east of PR have also dealt with military bombing activities, whose ecological impacts include coral colony fragmentation and permanent bio-construction damage (*Hernández-Delgado et al., 2014a*). The combination of global and local stressors can have a compound effect on the persistence and resilience of coral reefs. Nevertheless, global stressors are challenging to address directly (*Frieler et al., 2013*; *Schaefer et al., 2014*). It is therefore necessary to minimize local stressors and prioritize site-specific management actions (*Ban, Pressey & Graham, 2014*). However, in the Caribbean, the performance and benefits of local management vary widely and hinge on context-specific enforcement capacities (*Gill et al., 2017*; *Steneck et al., 2018*). Hence, it is important to evidence the diverse outcomes of local challenges and management approaches (*Aswani et al., 2015*).

In the face of these multi-scale, diverse and synergistic stressors, effective coral reef management requires tools able to inform on and adapt to ecosystem processes (*Flower et al., 2017*; *Ford et al., 2018*). Ecological resilience is operationalized as the magnitude of disturbance that can be absorbed by an ecosystem before it changes to a different state (*Carpenter et al., 2001*). Such an event can be the regime-shift from coral to sessile organism dominance because of excessive nutrient inputs and over-fishing of herbivore species (*Norström et al., 2009*; *Graham et al., 2014*). The combination of local stressors on coral-dominated reefs with low-grazing rates can lead to an increase in algae cover (*Elmhirst, Connolly & Hughes, 2009*), and a tendency for massive reef-building corals to be replaced by more common, encrusting coral taxa (*Edmunds, 2010*). In the Caribbean, coral reefs are shifting from high coral cover to alternate states of varying substrate rugosity, live coral cover, grazing intensities and algae abundances, which in turn are leading to novel ecosystems (*Hughes, 1994*; *Hughes et al., 2013*; *Precht et al., 2019*). This general shift has been largely attributed to the population collapse of the sea-urchin *Diadema antillarum* (*Lessios, 2016*), overfishing of herbivorous fishes, and the synergistic impacts of this loss of top-down herbivory control (*Mumby, Hastings & Edwards, 2007*). Some emergent coral reefs can maintain, albeit simplified, the ecosystem functions behind provisioning ecosystem services (*Norström et al., 2009*; *Graham et al., 2014*), yet how these
reefs are changing in locations in-between "pristine" and "degraded" states is still relatively unknown (*Mumby, 2017*). Minimum recovery and even irreversible decline in coral cover indicate a persistent regime-shift, but the thresholds to differentiate local alternate stable states from major structural shifts in community composition remain a challenge (*Rogers & Miller, 2006*).

Meanwhile, the long-term resilience of coastal social-ecological systems depends on the adaptability of front-line ecosystems, such as coral reefs, and their services to human communities (*Hernández-Delgado, 2015*; *Link et al., 2017*). Under this context, managers need to understand what ecosystem functions are desirable, achievable, and most critical for local-scale conservation (*Bellwood et al., 2019*).

The inherent complexity of these ecosystems compels the integration of metrics able to represent site-specific processes (*Lam, Doropoulos & Mumby, 2017*). Benthic community structure based on species abundances has been a common metric to monitor coral reefs, and despite the need to quantify more resilience-based approaches, refined benthic community structure assessments are still necessary (*Flower et al., 2017*; *Ford et al., 2018*). This is because even small changes in species abundance can have powerful impacts on coral resilience (*Mumby, 2017*), and our capacity to manage reefs effectively may be limited by novel species configurations (*Bellwood et al., 2019*). However, without the proper combination of other metrics, detailed community parameters cannot quantify specific ecosystem processes affected by exposure to environmental drivers (*Lam, Doropoulos & Mumby, 2017*).

Spatial analysis is a fundamental approach to characterize coastal ecosystem resilience because it can integrate bio-physical (*Magris et al., 2016*) and human phenomena (*Koch et al., 2009*). It also becomes a useful approach when assessing socio-ecological vulnerability to anthropogenic stressors (*Thiault et al., 2017*). Ecosystem vulnerability is often conceptualized as a function of the exposure, sensitivity and adaptive capacity of the perturbed organisms or ecosystems (*Adger, 2006*). Following spatial reasoning, we defined exposure as the distance between a measured ecological parameter and a specific local stressor (*Maina et al., 2011*; *Arkema et al., 2013*; *McLean et al., 2016*).

This study characterized the current ecological state of coral reefs in Flamenco Beach of Culebra Island, a touristic beach facing increasing anthropogenic pressures. Flamenco's historical decline in live coral cover, from 64% in 1986 (±18.2 95% CI) to 14% (±7.2 95% CI) in 2013 (E. Hernández-Delgado, 2013, personal observations), suggests it is transitioning into a coral-depleted, novel reef. However, this historical data is limited to two spatially clustered sites and does not reflect the site's recent increases in recreational uses and adjacent coastal development (*Hernández-Delgado et al., 2017*). Therefore, the first objective was to analyze a comprehensive benthic community structure baseline of eight fringing, barrier, and shoreline reefs, and assess how these vary to local anthropogenic stressors. The second objective was to assess spatial hotspots for targeted conservation strategies. Overall, this will inform the identification of major human activities affecting natural coral reef functions and therefore support coastal decision-making (*Bunce, Pomeroy & McConney, 2003*).

## MATERIALS AND METHODS

Culebra Island is located between the eastern Greater Antilles and the northern Lesser Antilles of the Caribbean Sea, approximately 27 km east of PR. It encompasses a total area of 2,728 hectares and a cluster of 26 cays. The local climate is greatly influenced by the seasonal Atlantic tropical storms between June and November. This is the season in which most of the 1,120.3 mm of annual rainfall is received (monthly average 1987–2017; W. Kunke, 2018, unpublished data). Culebra does not have permanent rivers or aquifers, which has historically reduced coastal sedimentation. However, it does exhibit ephemeral streams and highly erodible clay soils, which can result in high sediment loading to critical coastal habitats (Ramos-Scharrón, Hernández-Delgado & Amador, 2012). The watershed of the highest peak in the island, Mt. Resaca (198 msl), partly drains into Flamenco Bay.

Flamenco Bay was chosen as the study location due to its diverse reef habitats and recreational popularity. It is a horse-shoe shaped bay facing north, open to the predominant swell direction and therefore a dynamic and heterogenous coastline. While the eastern area is characterized by a fringing reef formation, the center and western sections are composed of sand, algae turfs, and beach rock formations. The site has distinct reef zones, including a barrier reef, a spur and groove back-reef, and a deep channel accompanied by patches of shoreline reefs. All zones exhibit a mixture of linear reefs, aggregate reefs, pavement, and rubble seafloor, with differing benthic heterogeneity. The bay has an extensive hard coral cover, but algae dominate the biological cover on patchy structures, suggesting that reef flats, back reefs, and shoreline fringing reefs are in declining ecological health (Kågesten et al., 2015). The study area (3.04 km$^2$) is frequently used by recreationists, which facilitated observations from the reef users and focused efforts on the most accessible and potentially threatened reefs.

### Sampling design

A total of eight distinct reef locations were sampled across the bay during July–September 2017. These reef locations were selected through stratified sampling according to homogenous features in benthic maps. Locations were chosen not just for their inherent spatial variation but for their similarity in aggregate reef geomorphology and varying degrees of rugosity. Benthic ecological parameters were measured at these reef locations, each composed of five, 10 m length replicate photo-transects, for a total of 40 transects across the study area (Fig. 1). This length has been appropriate for a leveling of cumulative coral species in the nearby island of St. John (Rogers et al., 1994). The transects were haphazardly selected once a geographic positioning system (GPS) confirmed it fell under the desired benthic characteristics, including bathymetric contours between 3.0 and 4.0 m in depth. The beginning and end of each transect were georeferenced.

For each of the five replicate transects in each reef zone, 0.5 m$^2$ quadrats were laid on top of the measuring tape at two-meter intervals and photographed vertically with a camera, for a total of six images per transect. Posteriorly, 48 regular grid points over these photos were used to identify the benthic structure inside each quadrat to the species level, including the percent of live scleractinian, hydrocoral, octocoral, sponge, macroalgae, algal turf, crustose coralline algae (CCA), erect calcareous algae (ECA), cyanobacteria cover,

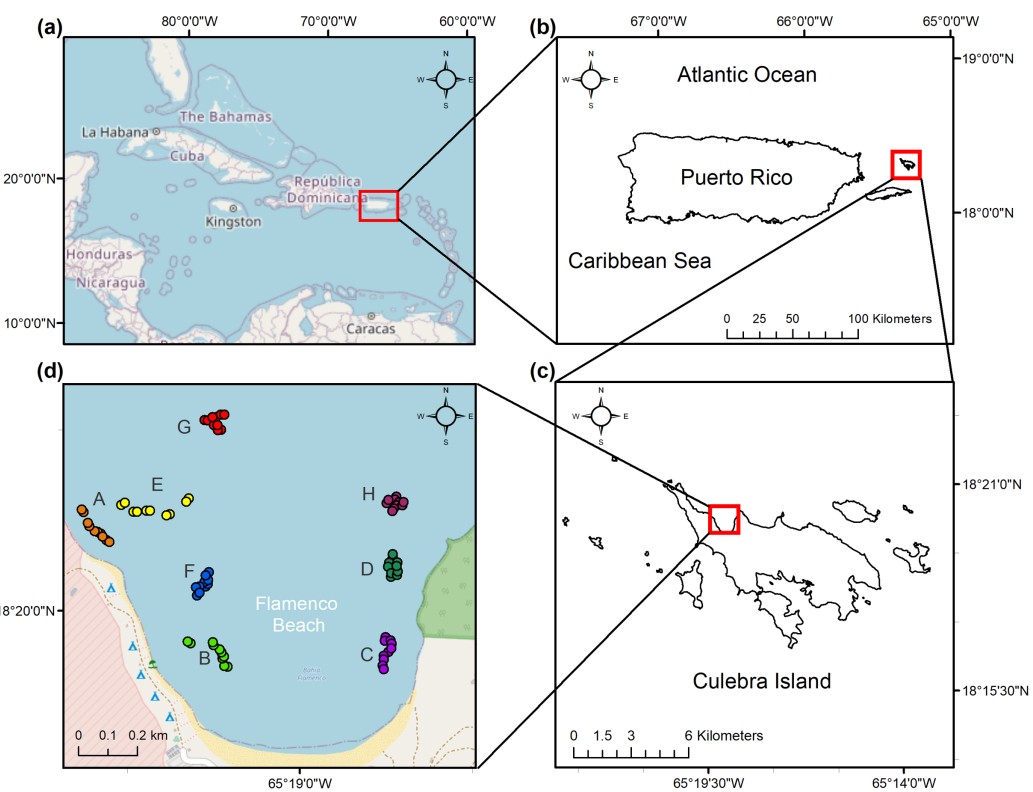

**Figure 1 Location of the study area.** (a) Caribbean Region. (b) Culebra Island in relation to the archipelago of Puerto Rico. (c) Flamenco Bay in relation to Culebra Island. (d) Beginning and end of the 10 m sampling transects as GPS waypoints, colored and labeled with a unique letter to represent each reef locality. Note that upper-case letters correspond to the reef sites sampled, while lower-case letter refer to the figure panels.

plus the percent cover of sand, pavement or rubble. Several guides were used to identify species (*Humann, Deloach & Wilk, 2002*; *Littler & Littler, 2000*; *Dueñas et al., 2010*; *Zea, Henkel & Pawlik, 2014*). Six common coral diseases (aspergillosis, white plague type II, red blotch syndrome and black, white and yellow-banded diseases), three predation types (damselfish, fireworm, and gastropod), and five categories for bleaching severity were also quantified to assess colony-level health (*Weil & Hooten, 2008*; *Bruckner & Hill, 2009*). Any coral colony <5 cm was treated as a coral recruit, including coral species with sexually mature small sizes, such as *Siderastrea radians* (*Irizarry-Soto & Weil, 2009*; *Hernández-Delgado, González-Ramos & Alejandro-Camis, 2014b*). Coral recruit density, coral species richness, and the coral species diversity (H'c) and evenness (J'c) were posteriorly calculated. The rugosity of each transect substrate was measured by placing a 2.0 m length chain with 1.5 cm links at each transect and comparing this chain's distance to the transect's horizontal length (*Rogers et al., 1994*). These measurements were ranked into a Rugosity Index (*Graham & Nash, 2013*). In addition, the population density of *D. antillarum* was quantified 2 m across along the length of each transect and classified into ranges (*NEPA, 2014*; *Rodríguez-Barreras et al., 2014*). Reef locations were classified according to their exposure to aquatic recreational use. Observations were carried on seven days across summer months of high tourist arrivals by patrolling the reefs between

11:00 and 15:00 (the visiting period of charter boats and touristic busses), quantifying aquatic recreationists and classifying their maximum frequency into a Recreational Index (Table S1). Furthermore, reef locations were classified as responding to nearshore or offshore dynamics, based on a 150 m distance threshold between the closest transect waypoint and the closest shoreline to it (measured in ArcMap 10.5) to interpret exposure from land-based human stressors (Table S2).

Variables were averaged for each transect ($n = 40$) across all locations. Benthic cover plots were created in the package ggplot2 in "R" statistical software (*Wickham, 2016*). All statistical tests were done through the Plymouth Routines In Multivariate Ecological Research (PRIMER) software v7.013 + PERMANOVA 1.0 (*Anderson, Gorley & Clarke, 2008*; *Clarke et al., 2014*). Taxonomic variance was characterized by similarity percentages routine (SIMPER) tests calculated through Bray–Curtis average similarities for each location, indicating which species are likely influencing the community structure (*Clarke et al., 2014*). The average taxonomic distinctness ($\Delta^*$) between all pairs of species in a sample was used to avoid the dependance of species richness on sample size and as a measure of phylogenetic diversity (*Clarke & Warwick, 2001*; *Clarke, Somerfield & Chapman, 2006*). Permutational dispersion (PERM-DIST) tests were also done to measure β-diversity (*Anderson, Ellingsen & McArdle, 2006*). Principal component ordination (PCO) was used to identify which benthic community components influenced spatial patterns based on fourth root-transformed species abundances.

A lack of a significant difference in benthic community structure among reef locations was tested using both one-way and two-way permutational analysis of variance (PERMANOVA), with location, distance from the shoreline, recreation index, rugosity index and *D. antillarum* density as main factors for 9,999 random permutations. Our balanced experimental design and the data's lack of normality suits the strengths and limitations of this test. One-way PERMANOVAs yielded the traditional Fisher's *F*-value, yet without assuming normal distributions (*Anderson, 2005*). Two-way tests were carried out in Bray Curtis dissimilarity space, a widely applied for biological assemblances (*Bray & Curtis, 1957*) to understand the interacting factors that most explained variances in the community structure. The location factor was nested in distance and tested as a random factor to broaden the inference to the population level. All other factors were un-nested and treated as fixed, in which inferences are limited to the experimental level. Following significant interactions, pair-wise tests were also performed. To account for environmental co-variates (amount of recreational visitors, rugosity and coral diseases), another matrix based on the benthic cover (rather than species) was also tested with PERMANOVA using the same factors, using the appropriate resemblance of Euclidian dissimilarities (*Anderson, Gorley & Clarke, 2008*).

A modified version of the Inverse Distance Weighting (IDW) spatial interpolation was used on 12 environmental variables to estimate unknown values based on sampled points and illustrate reef hotspots for targeted conservation strategies (*De Smith, Goodchild & Longley, 2015*). This was done using ten neighbors, without applying smoothing, and extracting them by mask to reef areas. Spatial autocorrelation provided a statistical measurement of spatial dependance, leading to a better understanding of the underlying
spatial nature of the ecosystem. We did not assume the distance-decay relationship is constant over space and adjusted the power value according to the nearest neighbor statistic to increase spatial predictive accuracy (*Lu & Wong, 2008*). All spatial interpolations were done using the Geostatistical and Spatial Analyst extensions in the ArcMap v.10.5 software (ESRI Corp., Redlands, CA, USA).

Lastly, a coral reef resilience index (CRRI) was applied to each location to summarize results to managers. The CRRI is composed of thirteen benthic variables grouped into coral, threatened species, and algal components. It was developed to enable a quantitative comparison from a single survey event between different reef types and an indication of the trend in health rather than only the current state of the reef. Therefore, it is a rapid assessment method to survey novel coral reef assemblages and a useful tool for managers and decision-makers for both small-and large-scale monitoring (*Hernández-Delgado et al., 2018*).

## RESULTS

### Benthic parameters across reefs

Without taking into consideration spatial variations, macroalgae exhibited the highest mean live benthic cover (22.8% ± 3.0% CI), while the combination of live scleractinian and hydrocoral cover (16.7% ± 3.1% CI) ranked as the second most widespread live cover type. See Table S3 for details on other cover types.

Despite being selected for their similarities in geomorphological structure, transect-averaged reef locations varied in benthic composition (Fig. 2). Scleractinian coral cover appeared homogenous across all locations except B, where its 4.8% cover contrasted with the 15.1–26.2% range of the other locations. However, even though some offshore locations exhibited higher coral cover, inshore location C stood out for its scleractinian dominance (Fig. S1). Octocoral cover was highest at offshore locations (F, G, H; ranging from 11.2% to 21.5%), and lowest at onshore reefs (6.7–8.4%), though location E was the exception. CCA followed this same spatial tendency, with highest values at offshore locations G and H (26.9%, 28.9%), and lowest at nearshore locations (3.8–14.6%). Macroalgae showed the opposite tendency as octocorals and CCA, with nearshore locations composed of the highest cover (21.4–33.5%) as opposed to offshore locations (10.4–24.8%). To a lesser extent, this was also the pattern with cyanobacteria, as nearshore locations had the highest covers (0.9–9.5%), and offshore locations slightly lower (0.9–5.5%) cover. (Fig. S2).

Herbivory, in the form of the long-spine urchin *D. antillarum* densities, did not vary remarkably across locations (0.1–0.3 $m^{-2}$), except for back-reef E, where the population density reached 0.75 $m^{-2}$. Furthermore, hard coral recruit densities did not surpass 2.0 $m^{-2}$, except for offshore location G (2.7 ± 1.0 $m^{-2}$). See Fig. S3 for details.

Rugosity was highest at nearshore locations B and D, with mean indexes of 1.7 and 1.8, respectively. Location B exhibited the distinct characteristic of deep crevasses running across the transect, which influenced this result. Meanwhile, low rugosity values were found mostly at offshore locations, notably front reefs G and H, and back reef E (1.3, 1.4 and 1.3, respectively). A One-way PERMANOVA test indicated significant differences

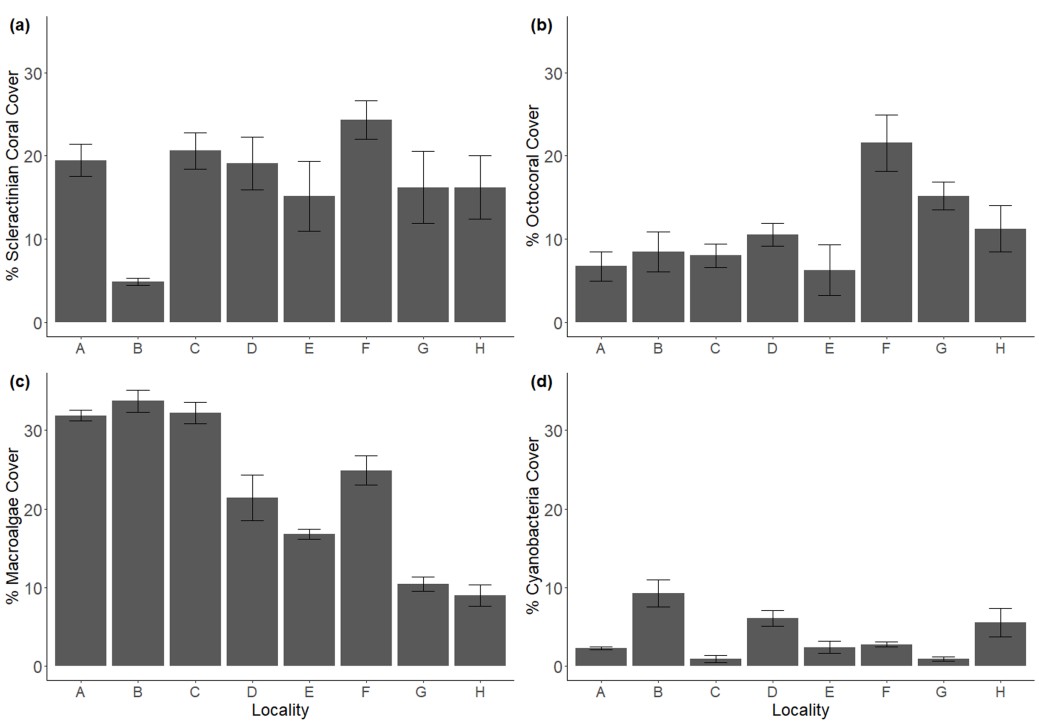

**Figure 2 Major benthic components in coral reef localities across Flamenco bay, shown in mean percent cover.** (a) Scleractinian cover. (b) Octocoral cover. (c) Macroalgae cover. (d) Cyanobacteria cover. Arrow bars represent the 95% confidence intervals. Refer to Fig. S2 for less abundant benthic components.              

among some locations (df = 7.39; Pseudo-*F* = 5.0895; *p* = 0.0007). A pair-wise test revealed significant relationships (*p* < 0.05) among reef locations. Specifically, inshore locations had higher rugosity than offshore zones (df = 1.39; Pseudo-*F* = 18.325; *p* = 0.0006), with notable differences (*p* < 0.05) between locations B and E, D and H, plus D and E.

## Biodiversity across reef locations

Massive reef-building coral species such as *Pseudodiploria strigosa* and *Orbicella annularis* were abundant at locations C, D and G. The encrusting *Porites astreoides* coral was less abundant but present in almost all locations. Meanwhile, brown macroalgae *Dictyota* spp. dominated across almost all the locations, while CCA was concentrated at specific transects of locations B, G and H. SIMPER tests ranked the macroalgae *Dictyota* spp. as the species contributing most to cumulative similarities at all the reef locations except location H, where CCA *Peyssonnelia* spp. out-ranked it by 17.39% (Table S4).

Hard coral species richness was highest at offshore reefs F, G and H with mean values of 14.0 ± 3.3, 12.8 ± 3.9 and 8.8 ± 3.2, respectively. Backreef location E had the lowest richness 5.4 ± 2.9. Meanwhile, coral species evenness (J'c) was similar across all locations, ranging from 0.98 to 0.99 and without statistical differences in any factors except the *D. antillarum* index (df = 3.36; pseudo-*F* = 5.772; *p* = 0.0277). A follow-up pair-wise test indicated that the only significance (*p* = 0.0330) was between the "critical" and "good"

**Table 1 PERMANOVA analysis of hard coral species richness (S) and diversity (H'c) across reef localities.** Analysis based on Log(x+1) transformed abundances of 57 species and Bray–Curtis dissimilarities.

| Factors | d.f. | Species richness (S) | | Diversity index (H'c) | | Component |
|---|---|---|---|---|---|---|
| | | Pseudo F | *P* (perm) | Pseudo F | *P* (perm) | |
| Distance | 1.38 | 0.2548 | 0.6485 | 0.25165 | 0.712 | Fixed |
| Rugosity Index | 3.36 | 1.1252 | 0.3459 | 1.3465 | 0.2636 | Fixed |
| Recreation Index | 4.35 | 2.0791 | 0.0952 | 1.9585 | 0.1034 | Fixed |
| *Diadema antillarum* Index | 3.36 | 1.4 | 0.2394 | 1.5451 | 0.2052 | Fixed |
| Location (Distance) | 6.32 | 6.0054 | **0.0005** | 5.2617 | **0.0006** | Random |
| Rugosity × Loc (Distance) | 18.20 | 2.3087 | **0.047** | 2.0362 | 0.1019 | Random |
| Recreation × Loc (Distance) | 6.32 | 6.0054 | **0.0007** | 5.2617 | **0.0008** | Random |
| *Diadema* × Loc (Distance) | 13.25 | 3.1003 | **0.0113** | 2.8005 | **0.0249** | Random |
| Distance × *Diadema* | 5.34 | 1.0833 | 0.3695 | 1.1168 | 0.3535 | Fixed |
| Distance × Recreation | 5.34 | 2.2138 | 0.681 | 2.0042 | 0.081 | Fixed |
| Distance × Rugosity | 5.34 | 0.82837 | 0.5404 | 0.91322 | 0.4893 | Fixed |
| Rugosity × *Diadema* | 10.29 | 1.4415 | 0.2042 | 1.5042 | 0.1806 | Fixed |

Note:
  Pseudo *F* statistics were calculated for each term using direct analogues to univariate expectations of mean squares. A conservative Type III sums of squares was used. *P*-values were obtained using 9,999 permutations under a reduced model, with each term contributing either a fixed or random component to the overall model. *P*-values in bold are below 0.05.

categories of this herbivory parameter. The largest variability in J'c was at C, due to the remnant dominance of *Acropora palmata*. Shannon's diversity index (H'c) and the average taxonomic distances ($\Delta^*$) both varied significantly ($p = 0.0429$ and $p = 0.0276$, respectively) across reef locations, suggesting the existence of coral phylogenetic diversity.

Species richness (S) of scleractinian corals was not significantly influenced by rugosity, *D. antillarum*, and distance from the shoreline (Table 1). However, S did vary significantly across reef locations ($p = 0.0005$), implying the adequacy of continuing to use S as an intuitive biodiversity measurement (Fig. S4). Pair-wise tests showed most of the combinations involving locations A and B were significant ($p < 0.05$). Variations in S were significant when considering interactions between location and the frequency of aquatic recreation ($p = 0.0007$). Moreover, H'c also was significantly different among reef locations ($p = 0.0006$), while all other factors were not. Location and recreation index ($p = 0.0008$), plus location and *D. antillarum* index ($p = 0.0249$) were also significant when tested against H'c.

## Multivariate patterns in benthic community structure

Principal component ordination analysis suggested that spatial variation only accounts for 28.0% of the total variation among benthic community structures in Flamenco Bay (Fig. S5). In locations D and E this variation was largely explained by the *O. annularis*, while other inshore locations, such as A and B, variability was explained more by brown macroalgae *Padina* spp. and *Dictyota* spp. Meanwhile, variation in offshore locations

**Table 2 Permutational analysis of variance (PERMANOVA) results of benthic community structure (236 species).** Analysis based on fourth root transformed abundances and Bray-Curtis dissimilarities, plus live coral, macroalgae and cyanobacteria based on square root transformation of % cover and Euclidian dissimilarities.

| Factors | d.f. | Component | Benthic community structure | | Live coral cover | | Macroalgae cover | | Cyanobacteria cover | |
|---|---|---|---|---|---|---|---|---|---|---|
| | | | Pseudo F | P (perm) | Pseudo F | P (perm) | Pseudo F | P (perm) | Pseudo F | P (perm) |
| Distance | 1.38 | Fixed | 3.9697 | **0.0001** | 0.84515 | 0.3712 | 15.504 | **0.0006** | 1.0293 | 0.348 |
| Rugosity Index | 3.36 | Fixed | 0.9554 | 0.577 | 1.2897 | 0.2888 | 2.5808 | 0.0664 | 1.6139 | 0.1776 |
| Recreation | 4.35 | Fixed | 3.5968 | **0.0001** | 2.6421 | **0.0434** | 7.4505 | **0.0004** | 0.99765 | 0.3739 |
| *Diadema* Index | 3.36 | Fixed | 1.644 | **0.0024** | 0.30943 | 0.8311 | 0.94652 | 0.4264 | 1.5125 | 0.2099 |
| Location (Distance) | 6.32 | Random | 3.368 | **0.0001** | 8.0592 | **0.0002** | 3.4613 | **0.0105** | 2.3539 | **0.0446** |
| Rug × Loc (Distance) | 18.20 | Random | 1.6768 | **0.0001** | 3.215 | **0.0113** | 1.9026 | 0.0967 | 1.7082 | 0.1825 |
| Rec × Loc (Distance) | 6.32 | Random | 3.368 | **0.0001** | 8.0592 | **0.0001** | 3.4613 | **0.0094** | 2.3539 | **0.0435** |
| *Dia* × Loc (Distance) | 16.22 | Random | 2.0492 | **0.0001** | 3.282 | **0.0097** | 1.274 | 0.2964 | 1.486 | 0.2559 |
| Distance × *Diadema* | 5.34 | Fixed | 2.0386 | **0.0001** | 0.35696 | 0.8901 | 4.0011 | 0.0058 | 0.9672 | 0.4464 |
| Distance × Recreation | 5.34 | Fixed | 3.6386 | **0.0001** | 8.5719 | **0.0001** | 7.5387 | **0.0002** | 1.006 | 0.3967 |
| Distance × Rugosity | 5.34 | Fixed | 1.5953 | **0.0008** | 0.7612 | 0.5883 | 4.292 | **0.0038** | 1.2102 | 0.2759 |
| Rugosity × *Diadema* | 10.29 | Fixed | 1.2218 | **0.0376** | 1.3486 | 0.245 | 1.6478 | 0.1333 | 1.4517 | 0.226 |

**Note:**
Pseudo *F* statistics were calculated for each term using direct analogs to univariate expectations of mean squares. A conservative Type III sums of squares was used. *P*-values were obtained using 9999 permutations under a reduced model, with each term contributing either a fixed or random component to the overall model. *P*-values in bold are below 0.05.

was mostly explained by *P. strigosa*, *Gorgonia flabellum*, and different CCA species. When interactions between factors were analyzed for the whole benthic community, all interactions with distance, location, rugosity, recreation and *D. antillarum* were statistically significant (Table 2). A clear separation among inshore and offshore reef locations was noticeable in the benthic community structure. Nearshore locations were largely dominated by sand, *Dictyota* spp. and *Padina* spp. and *O. annularis*. Offshore reefs were mostly dominated by CCA, non-reef building corals, and by zoanthid, *Palythoa caribaeorum*. The variation in benthic community structure was supported by PERM-DIST tests (a measure of β-diversity), which were significant among locations ($p = 0.001$) and distance to the shoreline ($p = 0.0131$). Furthermore, Metric Dimensional Scaling (MDS) plots of fourth root-transformed species abundances in a Bray–Curtis dissimilarity space of benthic averages showed how all reef locations were likely (95% CI) different from each other (Video S1).

The community structure did not exhibit significant differences across different benthic rugosities. This highlighted the dominating effect of distance from the shoreline when it interacted with rugosity, which was significant ($p = 0.0008$). The recreation index yielded significant relationships with benthic community structure ($p = 0.0001$) across all categories. The *D. antillarum* index also showed significant relationships with benthic community structure ($p = 0.0024$). Almost all reef locations fell under the "critical" and "poor" categories, and the few "fair" and "good" transects were clustered around offshore locations (Fig. 3).

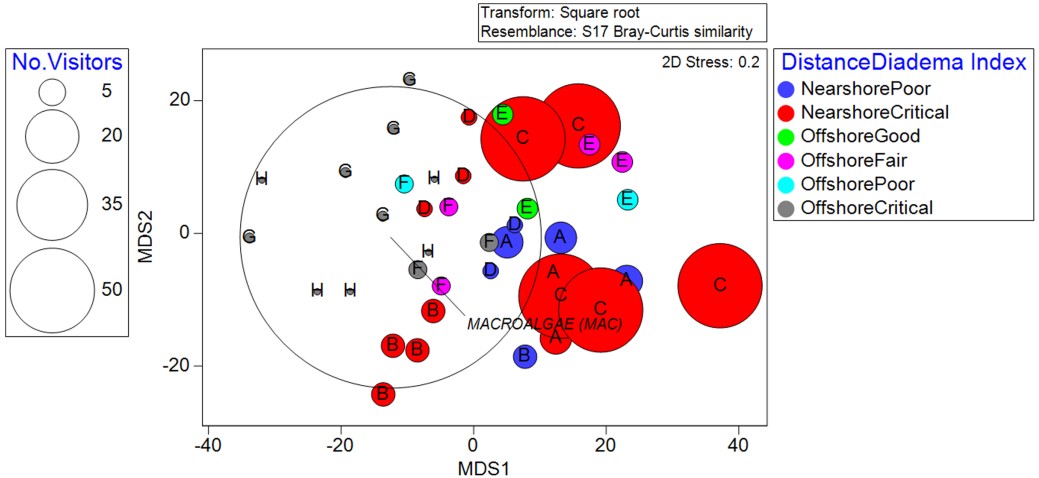

**Figure 3 Metric Dimensional Scaling (MDS) plot of transects grouped into localities and plotted with bubbles according to the quantity of observed recreationists.** Colors denote the classifications between distance from the shoreline and the *D. antillarum Index*. Overall, nearshore locality C stands out as high in human use and critical sea-urchin herbivory, explaining its high macroalgae cover.

## Relationships between biological cover and environmental co-variates

Live hard coral cover did not exhibit significant interactions in response to the distance from the shoreline, which is congruent with how reef-building coral species were abundant at both nearshore and offshore locations. For example, *O. annularis* was the dominant taxon in C and D nearshore locations, while *P. strigosa* dominated at offshore location "G" (Fig. S6). Rugosity was also not a significant factor affecting live coral cover. However, the variability in live coral cover could be explained as a function of the recreation index ($p = 0.0434$). The interaction between distance and recreation was also significant ($p = 0.0001$). This sustains that recreation exhibited a diminishing frequency gradient the farther the reef is from the shore, while coral cover increased the farther the reef is from shore. Pair-wise tests indicated that the significant relationship between live coral existed only when interacting with "low x moderate" recreational use (Table 3). Meanwhile, reef location was the most significant factor influencing live hard coral cover ($p = 0.0002$). Similarly, coral biodiversity varied significantly ($p = 0.0002$) in response to coral diseases (Table S5). The low coral recruitment densities across all locations did not vary significantly across coral cover values, suggesting even moderately high coral cover at inshore locations might not guarantee coral succession. Furthermore, there was a near absence of critically endangered corals *A. palmata* and *A. cervicornis* across all reefs (Fig. 4). *Acropora palmata* was mostly present at back-reef location E and other locations away from the shore. However, *A. cervicornis* was present at nearshore reef C.

Both distance and location were significant variables when interacting with macroalgae cover across reefs ($p = 0.0006$ and $p = 0.0105$, respectively). A pair-wise follow-up test confirmed 12 distinct location combinations had statistically different ($p < 0.05$) macroalgae covers. The brown macroalga *Dictyota* spp. was abundant in all locations, also evidenced through a PCO explaining 55.4% of the variance in benthic cover (Fig. S7).

**Table 3  PERMANOVA pair-wise combinations of ecological parameters in relation to the maximum frequency of aquatic recreation during peak hours.**

| Paired recreation categories | T-values | | | P-value | | |
|---|---|---|---|---|---|---|
| | Benthic community structure | Live coral % cover | Macroalgae % cover | Benthic community structure | Live coral % cover | Macroalgae % cover |
| Very Low × Low | 1.8191 | 1.6996 | 2.0231 | **0.001** | 0.1128 | 0.0614 |
| Very Low × Moderate | 2.0298 | 1.5966 | 4.3867 | **0.001** | 0.1146 | **0.0004** |
| Very Low × High | 2.4706 | 0.22445 | 4.4675 | **0.0003** | 0.8551 | **0.0003** |
| Very Low × Very High | 2.2299 | 0.29578 | 2.7205 | **0.0003** | 0.7937 | **0.0229** |
| Low × Moderate | 1.758 | 2.6156 | 1.0578 | **0.0002** | **0.0157** | 0.3034 |
| Low × High | 1.6308 | 2.3068 | 2.5799 | **0.0078** | 0.053 | **0.0091** |
| Low × Very High | 1.4702 | 1.9917 | 0.78274 | **0.0075** | 0.0753 | 0.4146 |
| Moderate × High | 1.5933 | 1.1362 | 2.0625 | **0.0005** | 0.2678 | 0.0553 |
| Moderate × Very High | 1.9254 | 1.4977 | 0.29852 | **0.0002** | 0.1451 | 0.7768 |
| High × Very High | 1.6576 | 0.52889 | 1.0249 | **0.0067** | 0.5998 | 0.4033 |

**Note:**
P-values in bold are statistically significant (<0.05).

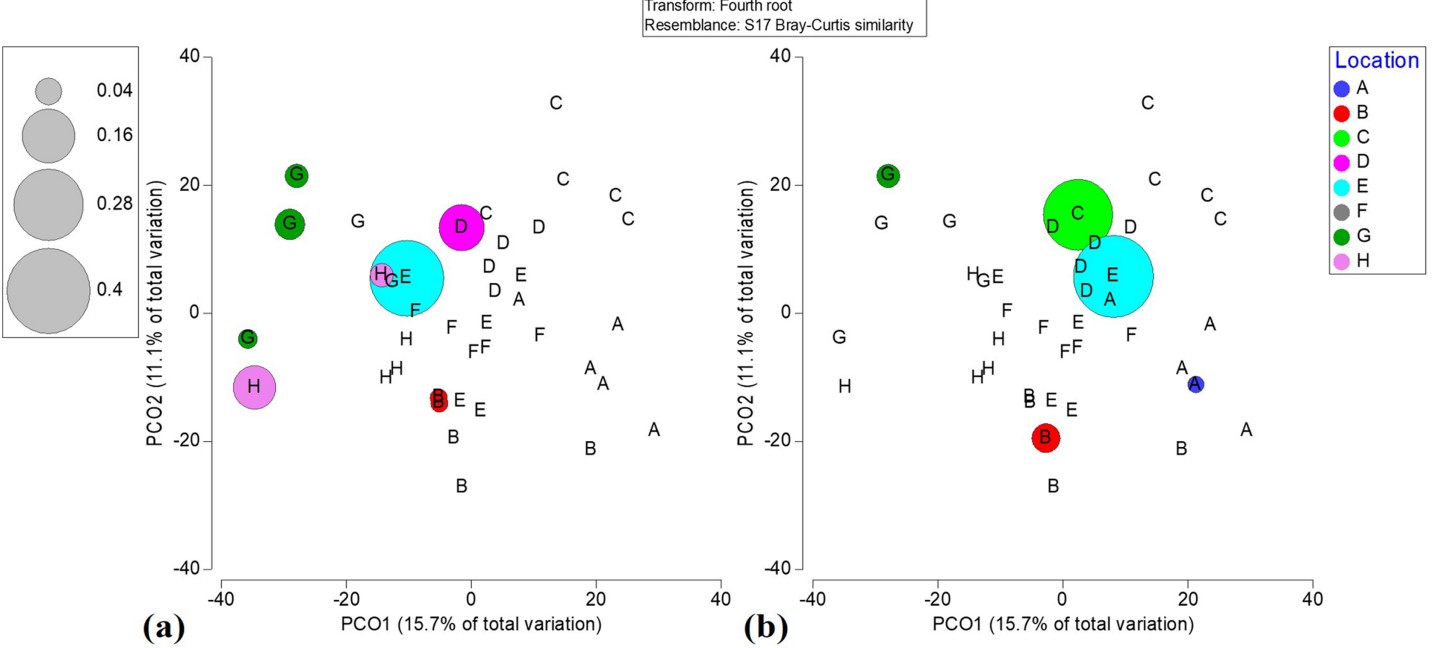

**Figure 4  Bubble plots based on PCO analysis describing the spatial patterns of critically endangered coral species among locations.** (a) *Acropora palmata.* (b) *Acropora cervicornis.*                         

Macroalgae abundances varied in locations from "very low" through "very-high" recreation densities (*p* = 0.0229). Pair-wise tests across 12 distinct location combinations confirmed that macroalgae were highest in nearshore reefs. The dominance of CCA, especially (*Peyssonnelia* spp. and *P. pachydermum*) on offshore reefs was evident through PCO analysis and contrasted to the clustered abundance of macroalgae community in the nearshore reefs (Fig. S8).

Cyanobacteria were most abundant at inshore reef B, near the recreationist swimmer's area, followed by offshore reef H. Variances in rugosity and *D. antillarum* indexes and their two-way combinations were not statistically significant. The combination of location and recreation did reveal marginally significant variance for the cyanobacteria community ($p = 0.0435$).

## Spatial analysis

Exploration of the spatial structure of sample points via the average nearest neighbor function revealed that the observed values (11.9 units) were smaller than those expected by a random dispersion (53.1 units), which generated a nearest neighbor ratio of 0.225 and indicated a clustering tendency. The global autocorrelation quantified through Moran's I covariance index equaled 0.923 ($p = 0.000005$ at a distance of 24.5 m), which indicated a very high positive spatial dependency and validated the appropriateness of performing IDW on tightly spaced samples over similar depths (*Li & Heap, 2011*; *Zarco-Perello & Simões, 2017*). Cross-validation analysis optimized the power value ($\alpha$) for IDW interpolations by both maximining the regression function between expected and observed values and minimizing its root mean squared error (Table S6).

Inverse Distance Weighting interpolations illustrated how ecological parameters associated with scleractinian corals were spatially variable yet revealed patterns useful for coastal managers (Fig. 5). The flat back reefs at location E for example exhibited low biodiversity, coral cover, and coral recruits, but also the last remaining populations of endangered Acroporids and highest disease abundance. Meanwhile, reef C exhibited some of the highest coral diversity, the highest macroalgae cover, and a low population of *D. Antillarum*. Additionally, sedimentation plumes originating from unpaved road erosion along the steep slopes of the eastern watershed were observed over inshore locations C and D (Fig. 6). The CRRI maps further supported how inshore reefs exhibit 'poor' ecological resilience, while offshore reefs still fell under a "fair" category (Table S7). None of the coral reef locations exhibited values indicative of high ecological resilience (Fig. 7).

## DISCUSSION

This study characterized the spatial variation in benthic community structure of differing shallow coral reefs and hypothesized that reefs with higher biodiversity, herbivory, recruitment would be in offshore habitats, far from coastal stressors, as partially observed in nearby islands (*Smith et al., 2008*). Overall, we confirmed the dominance of macroalgae (especially at several inshore locations), the low abundance of *Acropora* spp. and a high abundance of non-reef building corals across the ecosystem, trends that will most likely persist and shape the future of Caribbean reefs. While a homogenized species evenness points towards the coexistence of different coral species, we found statistically significant differences in benthic community structure with increasing distance from the shoreline and as a function of recreational use and sea-urchin density. This is consistent with other studies that showed spatial variation in benthic assemblages (*Fabricius et al., 2005*; *Hernández-Delgado et al., 2010*) and coral reef health conditions (*Smith et al., 2008*).

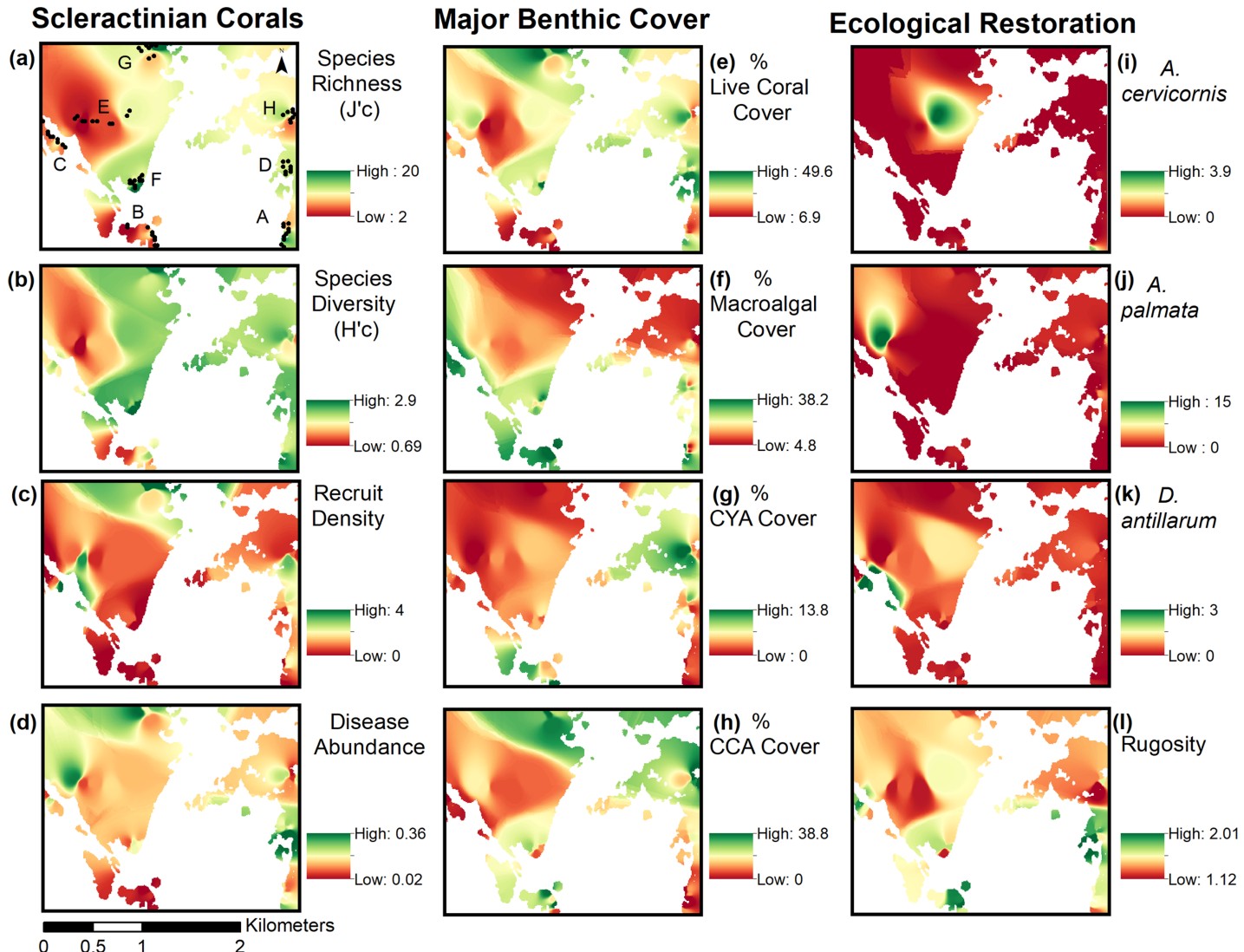

**Figure 5** **Inverse distance weighting (IDW) spatial interpolations of twelve ecological parameters in Flamenco Bay.** Parameters are grouped into three columns according to their characteristics. IDW layers are masked over aggregate substrate reefs, as mapped by *Kågesten et al., 2015*. Refer to Table S6 for input parameters and cross-validation results. Note that upper-case letters correspond to the reef sites sampled, while lower-case letter refer to the figure panels. (a) Species richness, (b) Species diversity, (c) Hard coral recruitment density, (d) Abundance of coral diseases, (e) Percent of live coral cover, (f) Percent of macroalgal cover, (g) Percent of cyanobacteria cover, (h) Percent of crustose coralline algae cover, (i) Abundance of *A. cervicornis*, (j) Abundance of *A. palmata*, (k) Abundance of D. antillarum, (l) Reef rugosity index.

The benthic composition of Flamenco's coral reefs was different across locations, suggesting the influencing role of oceanographic and anthropogenic factors on shaping the area's benthic community. For example, gorgonian-dominated hard bottoms were observed in offshore locations of high wave exposure. Also, different scleractinian coral species and functional groups dominated in close and far from shore. For example, columnar star coral, *O. annularis*, is a massive stony coral and an important reef-building species, and dominated at a shoreline location C, while brain coral, *P. strigosa*, was dominant at the farthest location from the shoreline, probably as a result of stronger surface circulation and larval supply.
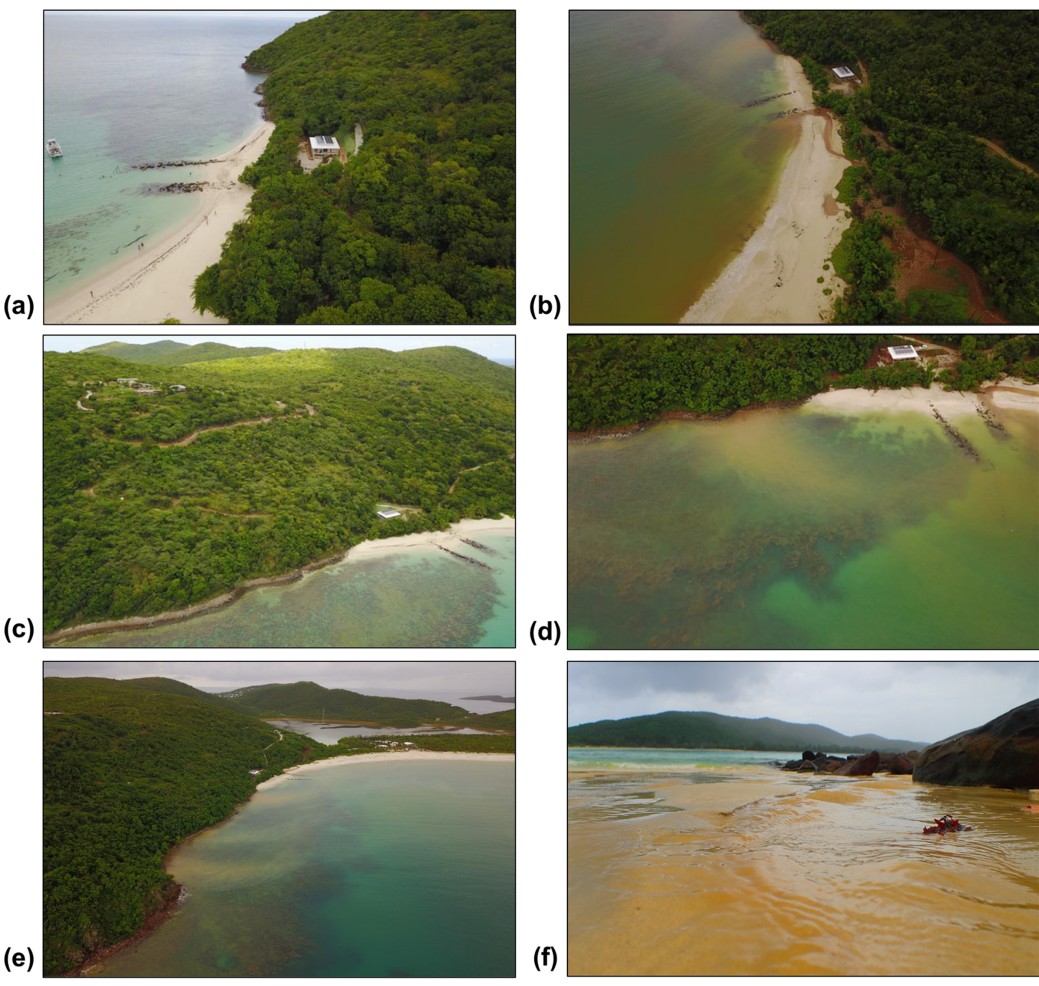

**Figure 6 Land-based source of pollution to Flamenco Bay from deforested hillsides and unpaved roads responding to unsustainable land use practices in 2017.** North-facing (a) and east-facing (b) views before the rainy season on 24th August. North-facing (c) and east-facing (d) views of sediment plumes on 7th November. (e) South-facing view of a suspended sediment plume dispersing over coral reefs. (f) Terrestrial sediment input adjacent to locality C. Photo credit: J. Acevedo.

Even though the rugosity of each location decreased as the distance from the shoreline increased, the reef's benthic composition did not vary significantly because of rugosity. This follows the regional trend that reef fronts and back reefs are flatter than fringing reefs, perhaps due to poor colonization in turbulent substrates (*Mumby, 2017*). A more nuanced statistical analysis showed how higher inshore rugosity does exhibit increases in macroalgae cover, which fits with trends in other Caribbean reefs (*Mumby, 2017*). Nevertheless, coral cover showed no significant spatial variation regardless of variation in rugosity, indicating that counter to the findings of other studies (e.g., *Graham & Nash, 2013*), Flamenco is not more coral-populated at locations with increased rugosity.

The near absence of *A. palmata* and *A. cervicornis*, which historically have dominated Caribbean reefs, in unison with the widespread abundance of ephemeral, non-reef building taxa (such as weedy coral *Porites astreoides* and brown macroalgae), implies changes in

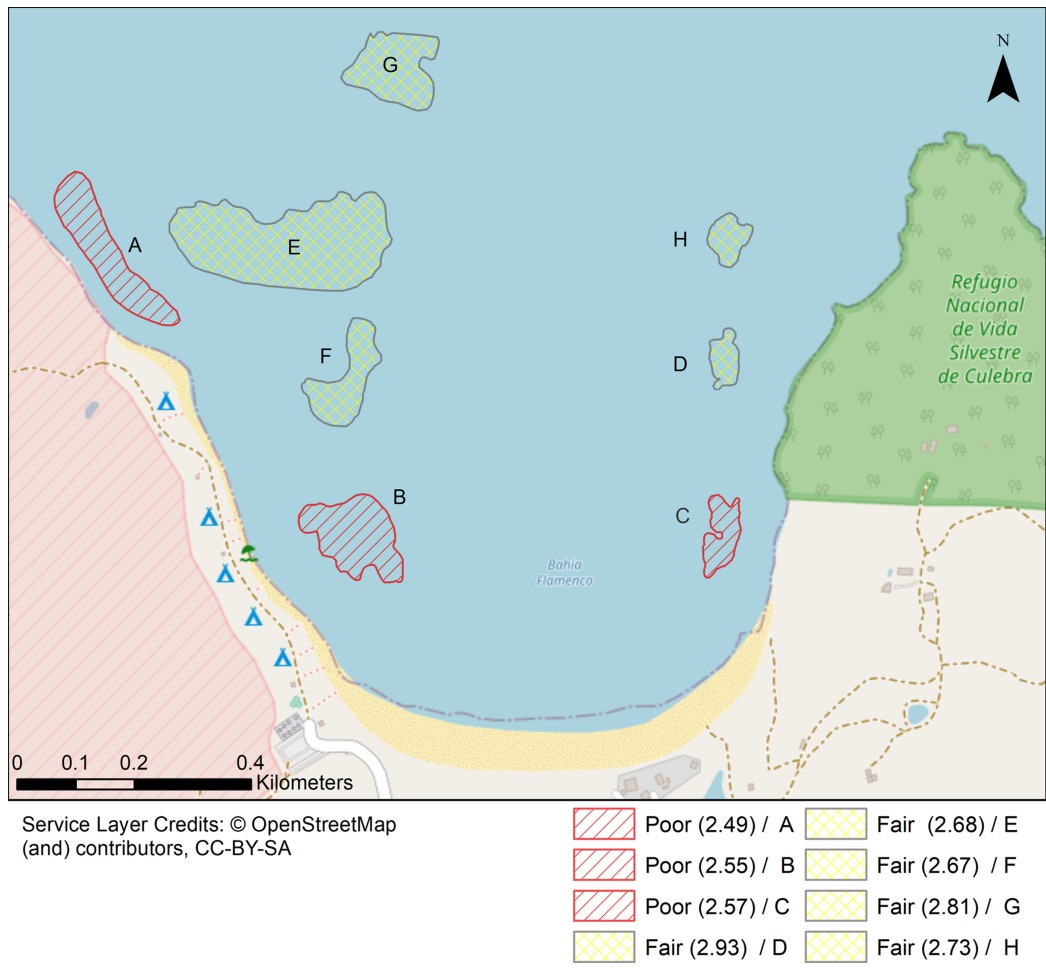

**Figure 7 Localities shaped according to a continuous geomorphic structure of aggregate reef and classified according to the Coral Reef Resilience Index (CRRI).** The CRRI is a consists of 15 indicators grouped into a either a Coral Index, Threatened Species Index and Algal Index, where the final mean value is deemed as very good (4.2–5), good (3.4–4.2), fair (2.6–3.4), poor (1.8–2.6) and critical (1–1.8). Refer to Table S7 for details. 

the benthic community in comparison to historical assemblages. Indeed, since 1986 live coral cover has significantly decreased while macroalgae cover has significantly increased in two reefs in the vicinity of contemporary localities C and D of Flamenco Bay (Fig. S9). This regime-shifting trend is consistent with previous studies across the Caribbean (*Hughes, 1994*; *McClanahan & Muthiga, 1998*; *Gardner et al., 2003*; *Miller et al., 2009*). Flamenco's widespread presence of thermal stress-tolerant and rapidly reproducing *P. astreoides* is consistent with its increasing abundance throughout the Caribbean and north-eastern Puerto Rico (*Green, Edmunds & Carpenter, 2008*; *Edmunds, 2010*; *Smith et al., 2013*; *Holstein, Smith & Paris, 2016*; *Soto-Santiago et al., 2017*). Meanwhile, the low abundance and centralized distribution of Acroporids across Flamenco may have potential adverse Allee effects on their population dispersion and connectivity across ecological spatial scales. IDW maps point to reef locations (E and C) where Acroporid restoration would assist in increasing rugosity and abundance of threatened species, thus contributing to

biodiversity conservation and coastal protection. This could be achieved by transplanting out-grown fragments from nearby coral nurseries to reefs locations with existing remnant populations using locally proven methods (*Hernández-Delgado, Mercado-Molina & Suleimán-Ramos, 2018*).

Meanwhile, *D. antillarum* densities were low throughout all of Flamenco's coral reefs, highlighting the critical state of long-spined sea-urchin herbivory. The highest (0.75 m$^{-2}$) densities were observed in back-reef E, which is consistent with previous findings associating higher populations to protected back-reefs (*Rodríguez-Barreras et al., 2014*). *Diadema antillarum* experienced widespread regional collapse over three decades ago and in recent years have shown moderate recovery (*Ruiz-Ramos, Hernández-Delgado & Schizas, 2011*; *Lessios, 2016*), partially due to limited successful recruitment (*Mercado-Molina et al., 2015*). Flamenco's reefs are not an exception in this trend, as *D. antillarum* does play a significant role in determining benthic community structure across increasing shoreline distances, but with overall low densities (<1 individual m$^{-2}$), below their capacity sustain critical herbivory functions (*Hughes, Reed & Boyle, 1987*). While nearby locations inside Culebra's no-take marine protected area have exhibited increasing densities (up to 2.75 individual/m$^2$ in 2016), Flamenco's low densities are more comparable to more heavily impacted urban locations in north-eastern Puerto Rico (*Rodríguez-Barreras et al., 2018*). Therefore, Flamenco can be classified as being in a critical state according to density metrics developed for the region (*NEPA, 2014*). These results build on the evidence found by a study during 2003–2004, which also sampled in Flamenco, where *D. antillarum* populations were characterized as critically low, especially since nearby locations exhibited the highest densities for Puerto Rico (*Ruiz-Ramos, Hernández-Delgado & Schizas, 2011*). We demonstrate *D. antillarum* populations have not recovered in Flamenco during the last decade. We recommend the propagation and restocking of *D. antillarum* in all of Flamenco's nearshore reefs following experimental methods developed for other Puerto Rican reefs (*Williams, 2016*), which have yielded promising results in reducing both macroalgae and encrusting red alga (*Williams, 2018*; *Olmeda-Saldaña, 2020*).

Flamenco's coral reefs are characterized by two main local stressors, harm from physical human activity (i.e., snorkeling) and sedimentation from unsustainable land-use practices. Different types of recreational activities occur across reef locations, with the highest frequency taking place at nearshore reefs easily accessible to individual snorkelers and where commercial charters provide un-guided snorkeling tours. Statistical analysis reveals that the frequency of aquatic recreational activity is correlated with significant variances in benthic community structure, specifically rises in macroalgae and cyanobacteria, plus decreased coral cover. Although the reef's degradation has not been causally linked to recreationists, we recommend minimizing this human stressor by establishing reef-specific recreational carrying capacities to both individual recreationists and touristic businesses (i.e., charter boats and equipment rentals), with a special focus to limiting the density of snorkelers. Additionally, efforts to educate best behavior practices in and around the reefs, such as signs and symbolic pledges when renting snorkeling and camping space, would minimize physical harm to corals and should be implemented by both businesses

and the local management agency (Authority for the Conservation and Development of Culebra (ACDEC)). Since coastal management is often more efficient when considering the behaviors and perceptions of stakeholders (*Diedrich, 2007*; *Oigman-Pszczol, Oliveira & Creed, 2007*), further strategies to mitigate coral harm from recreational activities should include visitor's willingness-to-pay for environmental management or perceived limits of acceptable change (LAC) (*Beharry-Borg & Scarpa, 2010*; *Schuhmann et al., 2016*). LAC has guided the zoning of reefs in similar touristic islands according to their ecological health and socially acceptable densities of <30 snorkelers (*Roman, Dearden & Rollins, 2007*), while reef user satisfaction, stewardship and capacity-building have been recommended as key socio-economic indicators of coral restoration (*Hein et al., 2017*). A participatory and transparent process should guide the implementation of reef zoning in Flamenco's reefs, to foment enduring local stewardship.

While recreationist's chronic impacts to corals need to be urgently minimized, unsustainable land-use also needs to be addressed. Unpaved roads have been characterized to be the highest contributor to erosion and a significant land-based source of pollution in Culebra Island (*Ramos-Scharrón & LaFevor, 2016*). Interpolation maps illustrate how nearshore reefs of high scleractinian abundance (notably *O. annularis* and *A. cervicornis* at reef C)*, overlap with the areas of high recreational densities and with areas close to recurrent turbid runoff outflows. Sediment particles inhibit larvae settlement and reduce coral diversity (*Risk & Edinger, 2011*), which may explain how reefs closer to this stressor exhibit some of the highest macroalgae covers and lowest recruitment densities in Flamenco. Terrigenous sediment has been shown to stress *O. annularis* colonies in Puerto Rico (*Acevedo, Morelock & Olivieri, 1989*) and both this species and *A. palmata* are sensitive to particle accumulation (*Rogers, 1983*). Although this study cannot determine a cause and effect relationship between sediment loads and declining coral health, this has been correlated for other nearshore coral reefs in Culebra (*Otaño-Cruz et al., 2017*; *Otaño-Cruz et al., 2019*).

Both local threats can be mitigated by addressing the coastal management deficiencies. For example, development projects in Culebra and touristic businesses in Flamenco must receive an endorsement from ACDEC, but this agency is underfunded often leading to failures in enforcing regulations effectively (*Johnston, 2003*). Thus, to effectively implement the restoration strategies outlined above, we recommend a co-management arrangement between ACDEC and local, community-based environmental organizations, which has been shown to lead to more successful outcomes in other coral reef socio-ecological systems (*Cinner et al., 2012*), including in Culebra's no-take marine protected area and the two decades of community-based coral restoration inside it (*Hernández-Delgado, Rosado-Matías & Sabat, 2006*; *Taylor, 2013*; *Hernández-Delgado et al., 2014c*). This approach is key in similar island contexts, where conventional approaches often suffer from weak compliance and enforcement (*Pollnac et al., 2010*; *Norström et al., 2016*). In the future, these results could be used to evaluate the socio-ecological resilience of Flamenco by incorporating reef user's spatial perceptions and behaviors (*Loerzel et al., 2017*) and aid in the implementation of ecosystem-based coastal management for Caribbean islands that depend on building sustainable touristic economies.

## CONCLUSIONS

This study found significant spatial variability in benthic community structure across differing shallow coral reefs of Flamenco Bay, Culebra Island. The difference between macroalgal-dominated communities in nearshore reefs versus dominant coral communities at offshore reefs, coupled with the near-absence of sea-urchins and Acroporid corals, point towards the elevated risk to coral species at reefs closer to the shoreline and coincide with their proximity to sedimentation pulses and increased recreational use. These spatial hotspots can serve as a guide to inform tourism management and ecological restoration efforts. Our findings also highlight the need to minimize recreational pressure in nearshore reefs through educational efforts and zoning of snorkeling densities. These management actions should be implemented through a co-management program between local agencies and community organizations. Governance of reefs key to the livelihoods of local communities, as those in Flamenco Bay are to Culebra, should be framed as an urgent need and opportunity for cross-sectorial partnerships. These findings have implications for coral reef management in other small tropical islands with increasing local threats, such as touristic development, rapid land-use changes, and poor governance, combined with global sea surface warming and sea-level rise.

## ACKNOWLEDGEMENTS

Appreciation goes to P. Gómez, G. Rosa, and K. Santana for sampling assistance and A. Montañez, A. Otaño, S. Suleimán, M.A. Lucking and M. Considine for their continued collaborative and logistical support. We thank William Kunke for providing his rainfall database. We give special thanks to Professor Marti J. Anderson at Massey University for her assistance in advising our statistical design.

### Funding

This work was supported by National Science Foundation (NSF), through the Center for Applied Tropical Ecology and Conservation at the University of Puerto Rico, under grant HDR-0734826. The funders had no role in study design, data collection and analysis, decision to publish, or preparation of the manuscript.

### Grant Disclosures

The following grant information was disclosed by the authors:
National Science Foundation (NSF).
Center for Applied Tropical Ecology and Conservation at the University of Puerto Rico: HDR-0734826.

### Competing Interests

The authors declare that they have no competing interests.

## Author Contributions

- Nicolás X. Gómez-Andújar conceived and designed the experiments, performed the experiments, analyzed the data, prepared figures and/or tables, authored or reviewed drafts of the paper, and approved the final draft.
- Edwin A. Hernandez-Delgado conceived and designed the experiments, analyzed the data, authored or reviewed drafts of the paper, and approved the final draft.

## Data Availability

All raw measurements are available in the Supplemental Files.

## Supplemental Information

Supplemental information for this article can be found online at http://dx.doi.org/10.7717/peerj.10080#supplemental-information.

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
