# Peer review of "Spatial benthic community analysis of shallow coral reefs to support coastal management in Culebra Island, Puerto Rico"

_PeerJ, doi:10.7717/peerj.10080_

## Round 0.1 · original submission · Major Revisions

This is an excellent research study with sound statistical methods to support conservation and management of an important place. I visited Flamenco Bay in 2003 and reading your ms brought back many wonderful memories.

I agree with Reviewer 1 that major revisions are required, including:
- Removal of superfluous information in the Results section and moving some text to Methods and Discussion.
- English grammar and spelling improvements are requested throughout. This includes additional sections in the paper to those identified by Reviewer 1.
- Clarify terminology: e.g., 'vulnerability' vs. 'exposure', 'risk', 'susceptibility', etc. These terms are often misused, confused, or used interchangeably in the literature. See Arkema et al., 2013 SI for an example of how you could define these terms up front in the context of coral reefs and stressors of Culebra (p. 5): http://msp.naturalcapitalproject.org/msp_concierge_master/docs/Arkema_etal_2013_NCC_SI.pdf
- Missing Biggs et al., 2012 in the References section.

I look forward to seeing the next version.

Reviewer 1 ·

Basic reporting

The writing needs some work and some of the references could be updated.
The manuscript is very long and could be considerably shortened. Analyses and results could be better tailored to objectives.

Experimental design

basic design is fine, and data is extensively analysed. Research questions could be better defined. Description of methods, particularly the statistical analyses needs to be clear and concise

Validity of the findings

replication is fine though this is a study looking at spatial differences, not temporal. Hence it is inappropriate to discuss findings with respect to change in benthos

Additional comments

This study assesses spatial variation in benthic reef communities within a bay subjected to recreational activity by tourists and terrestrial run-off. The authors show that there are spatial differences in the composition and diversity of benthic communities and relate these to indices of recreational activity, rugosity, distance from shore and urchin abundance. The analysis of data is extensive, though some clarification about statistical design, why analyses were performed and how they relate to study objectives is required. I also felt the results were very descriptive, somewhat repetitive and could be shortened considerably by focussing on the most relevant findings and deleting interpretive/ comparative text that references other studies and is better suited to the discussion.
The authors also need to be careful when interpreting their data which is from a single point in time. Hence it is difficult to know how these benthic communities may have looked in the past and what change these reefs have undergone. Finally, the manuscript needs to be carefully edited to remove ambiguous statements and correct grammar. I have provided some guidance on the first page of the introduction but this needs to be done throughout the manuscript.
L24 why would reefs closer to recreational stressors have higher diversity and herbivory? Similarly, the statement about urchin propagation and restoration L35-37) is not supported by information presented in the abstract.
L50 the Riegl reference is >10 years old and mass bleaching events since this paper was published have caused widespread loss of coral reefs. See Hughes et al 2018 Science 359:80-83
L55 the Wernberg et al 2016 Science? reference relates more to temperate reefs and if anything an increase in tropical species (such as corals) at the expense of macroalgae. Note it is also not in the reference list.
L59 you probably need to be explicit about stressors form recreational activity e.g. breakage of corals from snorkellers?
L63 delete ‘through reef craters’. It is nor the craters that cause damage to corals. Or is it?
L74 management ‘is’ critical
L80 The combination ‘of’ local stressors
L111-117 this is a long sentence that covers several issues. I suggest breaking into multiple sentences
L127-128 without a temporal data set that records change in benthic assemblage over time it is difficult to determine if a regime shift has occurred.
L131 I think you need to be clear about what you are assessing when discussing vulnerability. Is this vulnerability to anthropogenic stressors in general, climate change, high turbidity and sedimentation, or tourist activities? The vulnerability of a benthic community will depend on the stressors that you are assessing it against, and appropriate indictors of these stressors should be chosen accordingly using information from the literature to support your decisions.
Perhaps it is best to keep objectives fairly simple and achievable ie describe benthic communities and assess how these vary with respect to distance from shore, tourist densities. Relationships between benthic composition and the other two variables: urchin densities and rugosity are difficult to interpret as its impossible to know if these factors influence the benthos or are a consequence of the benthos, especially rugosity
L132 this will facilitate the ‘identification of’ the major human activities….
L135-139 what is the basis for this expectation? Also how do you know that the interaction between stressors is synergistic. This paragraph does not add very much to the introduction and could be deleted.
L153 is there more recent rainfall data than 1938-1972?
L184 and 188 are transects randomly or haphazardly placed?
L196 is there a reference for this measure of coral health.
L197 how is coral recruitment measured. What do you consider a coral recruit?
L204 if one of the aims of the study is to assess vulnerability then I think the methods need to be clear about how this will be done rather than referring to other studies.
Methods
L215 why is rugosity tested with ANOVA and other variables tested using PERMANOVA
In general, there is unnecessary information in the methods describing what an analysis does (e.g. L245-251, though number of permutations is needed as this can vary among studies) but information on how it is applied for this study is lacking.
There are several statistical tests used (ANOVA, PERMANOVA, PCO, SIMPER, IDW) and I found it difficult to follow what the response variables and factors were for these analyses. For example I don’t know how urchin density, location, distance from shore, rugosity and recreation index can all be factors in a one or two way PERMANOVA? Are some of these co-variates or are these multiple analyses. I wondered if all of these analyses are necessary to address the aims and if the analytical approach could be consolidated to make the manuscript more succinct.
I also think it may be better to use urchin density, rugosity, recreational activity and distance from shore as continuous variables in analyses rather than categorical. The authors should also clarify how distance from shore is measured. Is it distance to nearest shoreline, or the beach
Results
L296. This section is very long and I’m not sure that all subsections are necessary. For example, the section on overall benthic cover is very descriptive and it would be easier to just describe where there are significant differences in benthic variables, referring to figures as necessary. There are also sections when comparisons are made to published papers that are better suited to discussion e.g. L328—330
L357 is this dissimilarity between transects or locations?
L376 this opening sentence is awkwardly phrased. Perhaps better to say something like Species richness of benthic communities was not significantly influenced by rugosity, recreation….
More importantly, if there is a significant interaction between factors then the analysis should be interpreted at this level.
I’m also confused as to why there are interaction terms in Table 1 and 2 (which is labelled Table 1) and then terms that suggest a combination of two variables?
L440-514. This level of detail for correlations between benthic variables is very detailed and could be deleted, removed to supplementary files or shortened. Much of it is speculative, or contains text better suited to the discussion (e.g. L502-507) and does not add very much to what a description of the ordination in supp figure 6 would achieve. Indeed, this supp figure better summarises the data than fig 4 and 5 and could replace these in the main text.
L524-532 Fig 6 is a really nice summary of all the pertinent findings. However I don’t think this is the place to make suggestions about where restoration may be appropriate (L528).
L534-543 the description of the resilience index belongs in the methods, whilst interpretation of their findings is for the discussion. You should also explain how this aligns with your objectives in the methods.
Discussion
L550-555 the trends observed here are spatial not temporal. So it is inappropriate to suggest depletion of Acropora or trajectories through time based on a single assessment of benthic communities.
L584-586 Why would rugosity be positively correlated with macroalgal cover?
L592 has there been a loss of rugosity? The study shows spatial variation in rugosity
L613 here and elsewhere try to avoid overuse of emotive words or phrases like ‘alarming’ or ‘worrisome’.
L624 The presence of A palmata and A cerviconis skeletons certainly indicates that these corals were once present but without temporal data you con not really infer that there has been ‘expanding’ abundance of ephemeral corals and brown algae.
L651-654 This is a potentially important point. Though I’m not convinced that it is clearly illustrated in the results or analysis. If true it is somewhat lost in the other superfluous results and could be better presented.
L659-663 Its not clear to me how monitoring and analyses presented here would reduce sensitivity of benthic communities.
L690 I’m not sure the study characterised sediment plumes
L693 is this effluent or terrestrial run-off?
L705-707 agree that sedimentation and turbidly are stressors on some corals and can have a detrimental effect on their abundance/cover. But coral diversity is high close to shore (L703-4) which is somewhat inconsistent with this interpretation of findings?
L716-718 I’m not sure why the authors suggest some functional groups be omitted from future monitoring. Because these groups are low in abundance now does not mean they will be low in the future. This decision on which variables to included should be based on what are important indicators of benthic health and if they are sensitive to local stressors. Similarly, the basis for other recommendations for monitoring are not clear.
L725-728 the authors need to be careful when suggesting herbivory is low due to low abundance of urchins. As they note in the previous sentence there are other components of the faunal assemblage that are important for herbivory that have not been considered in this study.
L735 discussion about what locations are best suited to restoration is speculative, somewhat off-topic and could be reduced to a couple of sentences or deleted.
L747 similarly recommendations for making the bay a MPA are not well founded, off-topic and could be deleted
L805 what are these indicators? Are they synergistic? It could be argued the benthic community is less vulnerable if it has shifted towards coral species that are more tolerant of high turbidity and sediment loads.

Reviewer 2 ·

Basic reporting

Summary:

In this paper, the authors adopted a two-part approach to analyze the current status and future of coastal management practices in Culebra Island, Puerto Rico, the results of which can be applied to much of the Caribbean region. The authors conclude that reefs in Culebra Island are following the general trend seen in the greater Caribbean towards a degraded ecological state and a shift towards macroalgae-dominated and non-reef building coral-dominated benthic communities. Further, they list the critical state (and lack of recovery) of D. antillarum, reef flattening (a decrease in rugosity), and declines in coral cover as further threats facing Caribbean reefs of the future. These impacts leave reefs surrounding Culebra Island more vulnerable to anthropogenic and natural stressors such as tourism, pollution, hurricanes and sedimentation.

Following this analysis, the authors provide a detailed and interdisciplinary framework for how coastal management strategies can be improved to support conservation goals. These include specific strategies to integrate community stakeholders and local agencies into collaborative conservation efforts, such as the community-led enforcement of shallow-reef area No Take Zones to support herbivorous fish populations and build local economies. They recommend several key management actions that should be put into place to increase reef resilience, such as a citizen-science program to propagate D. antillarum and enhance the recovery of their populations.

Experimental design

The authors performed a thorough physical and biological characterization of shallow coral reefs neighboring a beach used heavily for tourism and recreational activities on Culebra Island to determine if different sites (moving from inshore to offshore) exhibited different characteristics. Photo-transects allowed for post hoc analysis of benthic community structure, including metrics such as reef rugosity, Diadema antillarum density, percent macroalgal and cyanobacterial cover, coral species richness and diversity, and substrate geomorphology. Across the eight reef sites, the authors found significant spatial variability in reef benthic community structure and composition. Specifically, shallow reefs closer to shore exhibited differences from reefs further offshore, which the authors associate with (at least in part) a decrease in anthropogenic stressors in the form of tourism and recreation moving further from the shore.

Validity of the findings

This paper is effective in that it provides a systematic analysis of the ecological status of a benthic coral reef ecosystem exposed to anthropogenic stressors and then uses this data to map out a series of management actions that can promote the recovery of coral reef ecosystems. The statistical analyses used to compare ecosystem characteristics at different sites were thorough and the resulting data support the authors’ conclusions. The literature is well-referenced and relevant. The authors do an impressive job at identifying relationships between variables and then predicting what ecological or natural process is responsible for driving them, such as the relationship between sea urchin populations and recreational human use. Finally, the authors provide informed predictions of the trajectory of coral reef health in the Caribbean and suggest revising the current system of monitoring by using novel health metrics that focus on location-specific resilience processes.

Overall, this study combines a characterization of a vulnerable and changing Caribbean coral reef environment with a novel perspective of how the environmental data can be used to inform and shape coastal management. This will be broadly useful to the field, helping to work towards a global “standard” for coral reef monitoring and the assessment of efficacy for conservation and coastal management strategies.

This manuscript should be revised to fix the few spelling and grammar errors present, of which specific comments and suggestions are left below. Following these minor edits, I believe this paper to be ready for publication in PeerJ.

Additional comments

Line 80: should read “the combination of local stressors…”

Line 110: should read “the inherent complexity of these ecosystems…”

Line 121: due to its seamless capacity to integrate…

Line 361: Change “has” to “had”

Line 471: change “these” to “this”

Line 474: should read “stood out was at location C…”

Line 475: the wording in this sentence is confusing, consider revising.

Line 499: add in “Cyanobacteria was most abundant at inshore reef B…”

Line 536: change to “quantitative comparison”

Line 597: change “specially” to “especially”

Line 786: “repetitive assessment of the interests”

---

## Round 0.2 · Minor Revisions

Please address English language issues and suggested improvements to the the discussion section raised by Reviewer 3. I agree that these are important revisions to increase impact of your paper.

No need to include a cover letter or response to reviewer in your resubmission. I will personally review the next version and make a decision. Thank you.

·

Basic reporting

ok

Experimental design

ok

Validity of the findings

ok

Additional comments

Dear Authors,

I have not seen the original submission, but based on your responses, you have addressed all the major points brought up the reviewers. However, there still is room for improvement in orthography and sentence construction, i.e. English.

I also think that the discussion is still a bit too long and should be more focused. What are the principal risk factors? What is their relative contribution? What recommendation would you make to prevent further degradation, or to assure recovery. Although there might not be clear answers or easy solutions/recommendations, it is important to propose mitigation measures, if you want your study to have impact.

Sincerely,

---

## Round 0.3 · accepted · Accept

The manuscript is greatly improved. I'm attaching the PDF of tracked changes for a few minor suggested edits to the Abstract and Discussion/Conclusion sections. Looking forward to seeing the published article. Great effort!